# Salinity and nitrogen source affect productivity and nutritional value of edible halophytes

**Tania Farzana, Qi Guo, Md. Sydur Rahman, Terry J. Rose, Bronwyn J. Barkla** *

Southern Cross Plant Science, Faculty of Science and Engineering, Southern Cross University, Lismore, NSW, Australia

* bronwyn.barkla@scu.edu.au

**Data Availability Statement:** All data are available in tables within the paper.

**Funding:** The author(s) received no specific funding for this work.

## Abstract

Saline agriculture may contribute to food production in the face of the declining availability of fresh water and an expanding area of salinized soils worldwide. However, there is currently little known about the biomass and nutrient/antinutrient accumulation response of many edible halophytes to increasing levels of salinity and nitrogen source. To address this, two glass house experiments were carried out. The first to study the shoot biomass, and nutrient accumulation response, measured by ICP-MS analysis, of edible halophyte species, including *Mesembryanthemum crystallinum* (ice plant), *Salsola komarovii* (Land seaweed), *Enchylaena tomentosa* (Ruby Saltbush), *Crithmum maritimum* (Rock Samphire), *Crambe maritima* (Sea Kale) and *Mertensia maritima* (Oyster Plant), under increasing levels of salinity (0 to 800 mM). The second experiment studied the effects of nitrogen source combined with salinity, on levels of oxalate, measured by HPLC, in ice plant and ruby saltbush. Species differences for biomass and sodium (Na), potassium (K), chloride (Cl), nitrogen (N) and phosphorus (P) accumulation were observed across the range of salt treatments (0 to 800mM). Shoot concentrations of the anti-nutrient oxalate decreased significantly in ice plant and ruby saltbush with an increase in the proportion of N provided as $NH_4^+$ (up to 100%), while shoot oxalate concentrations in ice plant and ruby saltbush grown in the absence of NaCl were not significantly different to oxalate concentrations in plants treated with 200 mM or 400 mM NaCl. However, the lower shoot oxalate concentrations observed with the increase in $NH_4^+$ came with concurrent reductions in shoot biomass. Results suggest that there will need to be a calculated tradeoff between oxalate levels and biomass when growing these plants for commercial purposes.

## Introduction

Halophytes are plants that have evolved to complete their lifecycle in high salt environments and are capable of tolerating a wide range of soil and water salinity levels [1–3]; many survive in seawater or substrates with even higher concentrations of salt [4, 5]. Halophytes use several specialized mechanisms (e.g., accumulation of inorganic salts, mainly NaCl in the vacuole, compatible solutes in the cytoplasm, and in some plants sodium (Na) accumulation in salt

**Competing interests:** The authors have declared that no competing interests exist.

glands, hairs and bladders) to allow them to tolerate salt, and to accumulate and sequester Na in the aerial parts of the plant. In some cases, they even show altered metabolism and the presence of specialized adaptive morphological features [6–8].

Halophytes are increasingly being used as salad ingredients owing to their crispy texture, the succulence of the leaves, and the salty flavor [3]. There is a market demand for edible halophytes, such as ice plant (*Mesembryanthemum crystallinum*), rock samphire (*Crithmum maritimum*), land seaweed (*Salsola komarovii*), oyster plant (*Mertensia maritima*) and quinoa (*Chenopodium quinoa*) in Asia and Europe, with a growing trend to cultivate for commercial production [3, 9]. In addition to and/or as a response to Na accumulation, halophytes accumulate high value nutritional components including primary metabolites such as osmolytes and scavengers of reactive oxygen species [10], amino acids, polyols and secondary metabolites including antioxidants (e.g. polyphenols, ß-carotene, ascorbic acid and ureides) [7, 11, 12]. Additional markets for halophytes with high nutritional potential may grow due to the pursuit of a healthier diet [3, 5].

Unfortunately, many of the halophytes used as raw foods contain anti-nutritional compounds including oxalates, nitrates, phenols, saponins and tannins [3]. For example, the halophytes ice plant, saltbush [5, 13, 14], and purslane (*Portulaca oleracea)* [15–17] accumulate high concentrations of oxalate in shoot tissue. A high intake of oxalates can have adverse health effects in vulnerable people, such as those at risk from kidney stones or low plasma levels of iron and calcium [18, 19]. Oxalate reduces blood calcium concentrations by forming insoluble calcium (Ca) oxalate in the body. In severe cases this induced Ca deficiency can lead to rickets, poor bone growth, and milk fever [20].

In order to establish halophytes as new food crops, and with the growing trend towards commercial cultivation [3, 9], it is necessary to optimize production systems to maximize yield and nutrient composition while minimizing accumulation of anti-nutrients such as oxalate and nitrate. Yield and nutrient quality of halophytes are influenced by salinity level and quantity of irrigation water, fertilization regime, harvest time, and the material harvested (young or old leaves, fruits, roots etc.) [3, 5, 21]. Concentrations of the anti-nutrient oxalate are impacted by both salinity level and the nitrate to ammonium nitrogen ratio ($NO_3^-$ : $NH_4^+$) of irrigation water [16, 22–24]. The halophytes purslane and saltbush accumulated oxalate when exposed to high levels of $NO_3^-$, whereas plants exposed to elevated $NH_4^+$ concentrations displayed a marked reduction in oxalate content [14, 19, 25, 26]. Unfortunately, $NH_4^+$ fertilization led to decreased biomass production in those plants. However, some studies suggest the deleterious effect of high levels of $NH_4^+$ on biomass production can be minimized by growing the plants in moderately saline conditions [14, 17, 26].

The aim of this study was to determine the effects of NaCl concentrations and $NO_3^-$ : $NH_4^+$ ratios in irrigation water on the growth, mineral nutritional quality and anti-nutritional compound levels (oxalate) of promising edible halophyte plants to optimize production methods for greenhouse horticulture. An initial glasshouse experiment investigated the growth (biomass) and nutrient uptake response of six edible halophytes to seven concentrations of NaCl in the irrigation solution. These results informed the salinity levels chosen for a second glasshouse experiment that investigated the effect of $NO_3^-$ : $NH_4^+$ ratios in irrigation water on the growth and shoot oxalate concentrations in two edible halophytes.

## Materials and methods

### Plant material

Plant species were selected with a focus on edible halophytes which have the potential to be incorporated into a greenhouse horticulture environment for commercial production. Seeds

and/or seedlings of six edible halophytes—ice plant, oyster plant, land seaweed, rock samphire, sea kale and salt bush—were sourced from commercial suppliers within Australia.

## Experiment 1 –Effect of salinity on growth and mineral nutrient accumulation of six edible halophytes

A glasshouse experiment was established to test the effects of seven levels of NaCl in the irrigation solution (0, 100, 200, 300, 400, 600 and 800 mM NaCl) on the growth and mineral nutrient accumulation of the six selected edible halophytes. The experiment was established in a randomized block design in a naturally-lit glasshouse at Southern Cross University (NSW, Australia; Latitude: -28.8173 and Longitude: 153.2987) with each halophyte species x NaCl concentration combination replicated three (rock samphire), four (ruby saltbush, sea kale and oyster plant) or five (ice plant and land seaweed) times depending on seed/seedling availability.

Seeds of ice plant and land seaweed were sown in experimental pots (plastic pots 155 mm diameter x 150 mm height) containing 2 L potting mix prepared with 1:1:1 peat moss, perlite and vermiculite. Oyster plant seeds were first placed in damp sand in a plastic bag and kept at 4°C for 2 weeks for vernalization. After vernalization, the oyster plant seeds were sown in the experimental pots at the same time as other seeds, as described for ice plant and land seaweed. Ruby saltbush and sea kale seeds were placed in small pots and kept in a growth chamber for germination, with 25°C and 20°C temperature maintained during the day (16 h duration) and night (8 h duration), respectively. After 3 weeks, the young seedlings were transferred to the experimental pots as described for ice plant and land seaweed. Rock samphire seedlings were purchased from a local market, and directly transplanted into the experimental pots. Two seedlings per experimental pot were kept for ice plant and land seaweed, whereas one seedling per experimental pot was maintained for ruby saltbush, rock samphire, sea kale and oyster plant. After the establishment of the seedlings (approximately 6 weeks after germination), the plants were treated with different levels of saline irrigation water (0, 100, 200, 300, 400, 600 and 800 mM NaCl) with replicate pots per treatment. Pots were watered to drainage on every second day with the appropriate irrigation water. A mean day/night air temperature of 26°C/21°C was maintained in the glasshouse with an average daily humidity of 48%.

## Experiment 2 –Effect of salinity and $NO_3^-$ : $NH_4^+$ ratios on ice plant and ruby saltbush shoot growth and oxalate/nitrate concentrations

An experiment was conducted on two edible halophyte species that had positive growth responses to 100 mM NaCl in the first experiment, ice plant and ruby saltbush, to investigate the interaction between salinity and $NO_3^-$ : $NH_4^+$ ratios on plant growth and concentrations of the anti-nutrients oxalate and nitrate in shoots. A hydroponic experiment was established in the glasshouse with conditions as per experiment 1 with a mean day/night air temperature of 26°C/21°C and an average daily humidity of 48%. Seeds of ice plant were sown in plastic pots (155 mm diameter and 150 mm height) containing 2 L potting mix, prepared by mixing 1:1:1 peat moss, perlite and vermiculite. Ruby saltbush seeds were placed in small pots and kept in a growth chamber for germination, with 25°C and 20°C temperature maintained during the day (16 h duration) and night (8 h duration), respectively. At 3 weeks after sowing, two seedlings of uniform size and vigor for each species were transferred into 5 L hydroponic containers wrapped with aluminum foil containing Yoshida nutrient solution [27].

For each halophyte species, a completely randomized design was adopted with five $NO_3^-$-N: $NH_4^+$-N ratios (1:0, 0.75:0.25, 0.5:0.5, 0.25:0.75 and 0:1) and three salinity levels (0, 200 and 400 mM NaCl). There were four replicate pots of each NaCl concentration x $NO_3^-$-N: $NH_4^+$-N

ratio combination. Nitrogen was supplied at 40 mg N/L. Macronutrient composition of the nutrient solution is presented in S1 Table in S1 File. Salt treatments were imposed 2 weeks after transferring seedlings into the nutrient solutions. For the duration of the experiment, nutrient solution in the containers was changed twice per week and the solution pH was monitored daily and maintained at 5.5–6.0 by addition of 0.5 M HCl or 0.5 M NaOH as needed.

## Measurements

In experiment 1, shoots were harvested at 70, 72, 80, 82 and 105 days after seed sowing for ice plant, land seaweed, oyster plant, sea kale and ruby saltbush, respectively, when the plants were at first flowering stage or just prior to flowering stage to ensure plants were all at the same level of maturity. Rock samphire was harvested 95 days after transplanting. Plant height and fresh weight of leaf and stem were recorded before all plant tissue was dried in drying room at 40˚C for 5 d, and then weighed and stored in an air-tight container prior to mineral analysis. Dried leaf and stem materials were combined and ground into a powder. A 0.2 subsample of powdered material was digested with 5 mL aqua regia and mineral concentrations in the digest were determined by inductively coupled plasma mass spectrometry (ICP-MS) in the Environmental Analysis Laboratory (EAL) at Southern Cross University, Australia. Mineral analysis was conducted on three biological replicates of each plant species x NaCl concentration combination.

In experiment 2, shoots (leaf and stem) were harvested at 5 and 6 weeks after transferring the seedlings into hydroponics for ice plant and ruby saltbush, respectively. Plant height and shoot fresh weight were recorded. Shoot material was then air-dried in a drying room at 40˚C for 5 d, and ground into a powder. A 25 mg subsample was transferred to a 5 mL Eppendorf centrifuge tube with 1.5 mL of MQ water. The solution was heated at 95˚ C in a water bath for 20 min and was then allowed to cool to room temperature. The solution was centrifuged at 15000 rpm for 20 min at 20˚C and 1 mL supernatant was then directly transferred to a 2 mL HPLC vial for oxalate and nitrate analysis. Two subsamples (analytical replicates) were prepared for analysis.

Oxalate and nitrate concentrations were determined using high performance liquid chromatography (HPLC) with a Shodex 1C SI-90 4E column in the Analytical Research Laboratory (ARL) at Southern Cross University, Australia. HPLC analysis was carried out using an Agilent 1260 HPLC System equipped with a vacuum degasser, quaternary pump, autoinjector, and diode array detector (DAD). The HPLC system was controlled using ChemStation software. Column temperature was set at 30˚C. A sample of 5 μl was chromatographed using 15 mM NaHCO3 as eluent at a flow rate of 1 mL/minute and the absorbance was measured at 280 nm. Peak areas were calculated using ChemStation software, and the resulting areas were compared to standard oxalate and nitrate curves to determine oxalate/nitrate concentrations.

## Statistical analyses

Analyses were conducted in the R software (version 4.2.1) framework. In experiment 1, biomass and shoot mineral concentration and content data were subjected to a two-way ANOVA fitting salinity (NaCl level) and plant species as factors. In experiment 2, shoot biomass, and oxalate and nitrate concentration data for each halophyte species were analyzed using a two-way ANOVA fitting salinity (NaCl level) and $NO_3^--N$: $NH_4^+-N$ ratio as factors. Means were compared using Fisher's LSD test at $P < 0.05.3$.

## Results

### Experiment 1

**Biomass.** For all species except ice plant, plant height generally decreased with increasing levels of salinity in the irrigation solution (Fig 1). For ice plant, plant height was greatest in plants treated with 100 mM NaCl in the irrigation water compared to the untreated plants, then progressively declined with increasing salt levels up to 800 mM NaCl (Fig 1). Leaf biomass and total biomass in ice plant, land seaweed, ruby saltbush and rock samphire increased with 100 or 200 mM NaCl in the irrigation water, but progressively declined with increased salt levels up to 800 mM NaCl (Fig 1). In contrast, leaf biomass and total biomass in sea kale and oyster plant were highest in the absence of salt treatment and progressively declined with increasing levels of salinity in irrigation water (Fig 1).

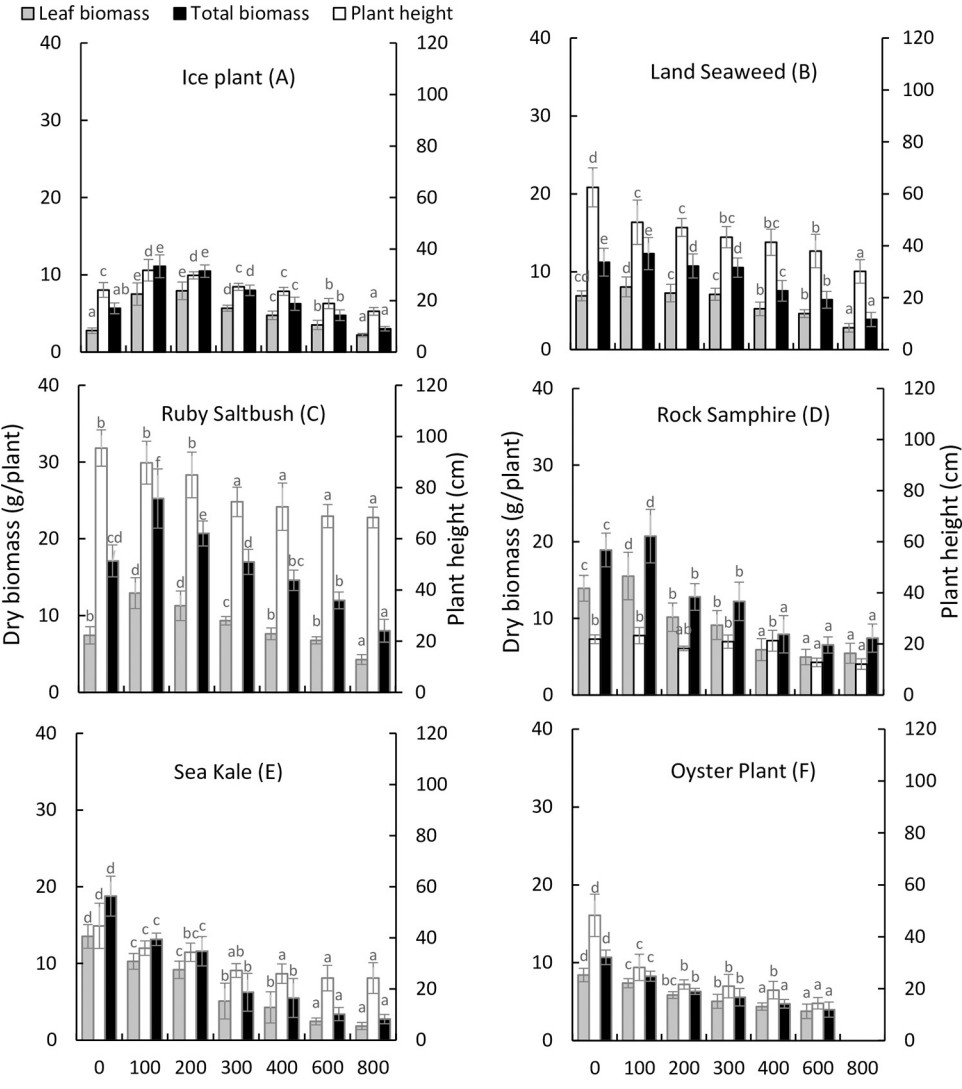

**Fig 1. Effect of NaCl concentration in irrigation water on edible halophyte height and biomass.** Values are means ± SD (n = 3 for rock samphire, n = 4 for ruby saltbush, sea kale and oyster plant and n = 5 for ice plant and land seaweed). Means that share a common letter are not significantly different at P < 0.05.

**Mineral concentrations in shoots.** The concentrations of nutrients in shoots of the halophyte plants under study reflected both nutrient uptake and the biomass production. All plants accumulated Na, with highest concentrations reached in ice plant and ruby saltbush at 600 mM NaCl treatment (225 mg/g and 155 mg/g, respectively) (Table 1). Shoot Na concentrations in land seaweed, rock samphire, sea kale and oyster plant ranged from 7–109 mg/g, 2–58 mg/g, 47–104 mg/g, and 34–103 mg/g, respectively, depending on the Na treatment (Table 1). A number of species in the study showed significant accumulation of Na in plants grown in the absence of added NaCl, ranging from 32 mg/g to 47 mg/g, highlighting the ability of these plants to hyperaccumulate Na from the soil. The exceptions to this were land seaweed and rock samphire which had concentrations of 7 and 2 mg/g Na, respectively, when grown in the absence of NaCl (Table 1).

In all halophyte species, leaf Cl concentration showed similar trends to those observed for Na (Table 1). Many of the species accumulated near maximum Na and Cl concentrations when treated with low concentrations of NaCl in the growth medium, with little or only small increases in shoot concentrations of these ions at higher levels of NaCl in the irrigation water (Table 1).

For the macronutrients nitrogen (N), phosphorus (P) and sulfur (S), there were significant differences in concentrations across species. While oyster plant, which overall had the lowest N levels compared to all other species, showed increased N concentrations when higher concentrations of NaCl (400 and 600 mM) were applied, the general trend for N was a decrease with increasing salinity (Table 1). However, the magnitude of the N decrease varied across species, with rock samphire showing a 10% decrease between plants treated with 0 mM NaCl compared to 800 mM NaCl plants and ice plant exhibiting a more than 50% decline across the same range (Table 1). Oyster plant had a significantly lower P concentration than all other species. Phosphorus concentrations declined in most species with an increase in salinity, with a > 70% decline in land seaweed from 100–800 mM NaCl (17.8–5.4 mg/g), while there was no significant decline in P concentration in sea kale with increasing salinity. The addition of NaCl (100 mM) caused a significant drop in S concentrations in most of the species except for sea kale and oyster plant, however, further significant decreases in S concentration with increasing NaCl concentration only occurred in ice plant and oyster plant.

Shoot concentrations of the macronutrient cations Ca, magnesium (Mg) and potassium (K) also differed across species and with NaCl treatment. Notably, the K concentration in rock samphire was 2.5 to 4.5-fold higher than for all other species (Table 1). Shoot K concentrations declined progressively with each increase in NaCl level for ice plant but increased significantly with treatment of plants with 100 mM NaCl in land seaweed and with 100 mM-600 mM NaCl in rock samphire (Table 1). All plants showed either little change or a decrease in magnesium concentration with increasing NaCl. One exception to this was a significant increase in Mg in rock samphire in plants treated with 800 mM NaCl. Shoot Ca concentrations were mostly unaffected by NaCl levels except in oyster plant where concentrations generally declined with increasing NaCl concentrations (Table 1).

Salinity treatment had no consistent effect on shoot iron (Fe) concentrations in most species. However, concentrations in rock samphire significantly declined from 0.166 mg/g (untreated) to 0.107 mg/g when 100 mM NaCl was applied (Table 1). Salinity did not cause any consistent increase or decrease in copper (Cu), zinc (Zn) or manganese (Mn) concentrations in most species (Table 1), with the exception of oyster plant Zn concentrations which significantly declined from 0.086 mg/g (untreated) to 0.045 mg/g when 100 mM NaCl was applied (Table 1).

**Table 1. Effect of salinity on mineral concentrations in edible halophyte shoots.**

| Plant | Salinity (mM NaCl) | Na | Cl | N | P | S | Ca | Mg | K | Fe | Cu | Zn | Mn |
|---|---|---|---|---|---|---|---|---|---|---|---|---|---|
| **Ice Plant** | 0 | 32.67b | 28.86b | 78.07a | 11.66a | 10.51a | 5.59abc | 9.56a | 47.97a | 0.29a | 0.031a | 0.152ab | 1.146a |
| | 100 | 190.35a | 92.32a | 60.67b | 9.27bc | 6.71b | 5.83ab | 7.91b | 42.22ab | 0.21b | 0.023b | 0.157ab | 0.604bc |
| | 200 | 208.40a | 94.98a | 47.87bc | 10.64ab | 4.79c | 4.23d | 5.65c | 34.62bc | 0.25ab | 0.024b | 0.134b | 0.879c |
| | 300 | 222.12a | 89.62a | 40.07c | 9.66bc | 4.41c | 4.45cd | 5.21c | 28.73cd | 0.27ab | 0.024b | 0.136b | 0.872c |
| | 400 | 217.13a | 94.88a | 37.57c | 8.35cd | 4.79c | 4.97bcd | 5.86c | 20.53d | 0.30a | 0.024b | 0.156ab | 0.962bc |
| | 600 | 225.14a | 96.73a | 33.40c | 6.98de | 4.86c | 5.32bcd | 5.70c | 23.69d | 0.31a | 0.032a | 0.161ab | 0.901bc |
| | 800 | 209.35a | 100.34a | 38.43c | 6.07e | 6.41b | 6.62a | 7.86b | 22.77d | 0.25ab | 0.032a | 0.178a | 1.056ab |
| | LSD | 44.46 | 24.10 | 14.64 | 1.52 | 1.51 | 1.17 | 1.59 | 9.06 | 0.0678 | 0.0048 | 0.0317 | 0.1577 |
| | Mean | 186.45A | 85.39A | 48.01B | 8.95A | 6.07BC | 5.29C | 6.82B | 31.51B | 0.269A | 0.027A | 0.153AB | 0.960A |
| **Land Seaweed** | 0 | 7.35d | 4.50c | 50.47a | 8.66bc | 8.32a | 4.41abc | 8.00a | 45.44b | 0.116ab | 0.035a | 0.115a | 0.642a |
| | 100 | 108.20a | 80.49a | 45.87b | 17.82a | 3.82b | 4.84a | 6.17b | 62.80a | 0.091b | 0.026b | 0.096b | 0.635ab |
| | 200 | 102.20ab | 79.56ab | 43.10c | 10.72b | 3.44b | 4.01cd | 5.12c | 42.08b | 0.097b | 0.025bc | 0.097b | 0.615ab |
| | 300 | 109.07a | 78.06ab | 41.83c | 8.77bc | 3.59b | 4.03bcd | 4.85c | 34.60bc | 0.100b | 0.026b | 0.103ab | 0.625ab |
| | 400 | 101.84ab | 75.11ab | 41.73c | 7.86bc | 3.33b | 3.76cd | 5.22c | 33.39bc | 0.102ab | 0.023c | 0.097b | 0.571b |
| | 600 | 90.07bc | 70.69b | 41.13c | 6.62c | 3.69b | 3.57d | 5.01c | 28.66c | 0.126a | 0.025bc | 0.104ab | 0.578ab |
| | 800 | 84.50c | 72.26ab | 38.40d | 5.39c | 3.98b | 4.66ab | 6.06b | 25.20c | 0.113ab | 0.027b | 0.111ab | 0.628ab |
| | LSD | 16.02 | 9.76 | 2.61 | 4.08 | 0.69 | 0.64 | 0.74 | 12.28 | 0.0255 | 0.0026 | 0.0176 | 0.0693 |
| | Mean | 86.18C | 65.81AB | 43.22B | 9.41A | 4.31CD | 4.18CD | 5.78B | 38.88B | 0.106C | 0.027A | 0.103C | 0.613B |
| **Ruby Saltbush** | 0 | 15.13f | 4.82c | 53.96a | 7.53ab | 5.07a | 2.46d | 4.03ab | 39.89a | 0.129a | 0.022a | 0.067a | 0.510a |
| | 100 | 163.12a | 78.09a | 45.93b | 9.64a | 2.62b | 3.78ab | 4.00ab | 23.51b | 0.061b | 0.024a | 0.084a | 0.527a |
| | 200 | 107.15e | 54.15b | 45.20b | 6.97b | 3.42b | 3.48abc | 4.35a | 19.19b | 0.110ab | 0.026a | 0.083a | 0.598a |
| | 300 | 142.82c | 65.90ab | 47.80b | 9.08ab | 2.87b | 2.99cd | 3.21b | 17.77c | 0.094ab | 0.020a | 0.069a | 0.490a |
| | 400 | 146.62bc | 66.36ab | 47.90b | 8.58ab | 3.47b | 3.22bc | 3.54ab | 15.59c | 0.093ab | 0.021a | 0.074a | 0.582a |
| | 600 | 155.54ab | 77.17a | 45.10b | 8.87ab | 3.56b | 4.15a | 4.31a | 15.60c | 0.127a | 0.025a | 0.097a | 0.681a |
| | 800 | 130.36d | 75.05a | 46.10b | 7.48b | 3.38b | 3.62abc | 3.82ab | 14.82c | 0.105ab | 0.021a | 0.085a | 0.524a |
| | LSD | 10.85 | 13.08 | 6.03 | 2.16 | 0.95 | 0.70 | 1.03 | 5.46 | 0.0532 | 0.0064 | 0.0401 | 0.2489 |
| | Mean | 122.97B | 60.22B | 47.43B | 8.31AB | 3.48D | 3.38D | 3.89C | 20.91C | 0.103C | 0.023B | 0.080D | 0.559B |
| **Rock Samphire** | 0 | 2.33c | 5.32d | 64.57a | 7.54a | 6.36b | 8.62c | 4.04b | 68.96c | 0.166a | 0.032a | 0.150b | 0.510b |
| | 100 | 11.41bc | 26.80cd | 61.70ab | 6.56bcd | 3.74c | 8.75bc | 3.28b | 88.56b | 0.107b | 0.029a | 0.184a | 0.523b |
| | 200 | 21.74b | 48.17bc | 59.77ab | 6.88abc | 4.06c | 10.24a | 3.47b | 100.53a | 0.090b | 0.029a | 0.200a | 0.558ab |
| | 300 | 48.24a | 64.54b | 59.67ab | 6.04d | 3.89c | 9.53ab | 3.29b | 98.47a | 0.086b | 0.026b | 0.185a | 0.539ab |
| | 400 | 55.09a | 95.30a | 61.57ab | 6.37cd | 4.07c | 9.55ab | 3.63b | 106.49a | 0.095b | 0.031a | 0.184a | 0.586a |
| | 600 | 58.27a | 70.55ab | 59.83ab | 6.49bcd | 3.81c | 9.04bc | 3.45b | 100.56a | 0.087b | 0.030a | 0.190a | 0.547ab |
| | 800 | 6.32c | 7.22d | 58.67b | 7.16ab | 22.64a | 9.58ab | 8.93a | 35.28d | 0.094b | 0.012c | 0.084c | 0.595a |
| | LSD | 10.54 | 26.31 | 4.91 | 0.68 | 1.21 | 0.90 | 1.14 | 9.35 | 0.0368 | 0.0031 | 0.0255 | 0.0583 |
| | Mean | 29.06D | 45.42B | 60.82A | 6.72C | 6.94B | 9.33B | 4.30C | 85.55A | 0.103C | 0.027A | 0.168A | 0.551B |
| **Sea Kale** | 0 | 47.84bc | 65.19a | 62.93a | 7.04ab | 18.21a | 10.09b | 9.42a | 41.57a | 0.104ab | 0.015a | 0.180ab | 0.681a |
| | 100 | 73.98ab | 74.43a | 64.20a | 7.11ab | 17.67a | 8.69b | 9.14a | 31.50a | 0.056b | 0.012ab | 0.131ab | 0.534ab |
| | 200 | 82.11ab | 76.89a | 63.77a | 8.75a | 15.59a | 9.82b | 8.69a | 33.07a | 0.081b | 0.013ab | 0.200a | 0.613ab |
| | 300 | 74.12ab | 83.91a | 67.00a | 6.37b | 17.36a | 9.85b | 9.29a | 37.29a | 0.056b | 0.012ab | 0.123abc | 0.619ab |
| | 400 | 100.85a | 82.76a | 63.53a | 6.74ab | 14.84a | 8.11b | 7.97a | 36.68a | 0.066b | 0.011b | 0.114bc | 0.579ab |
| | 600 | 104.82a | 84.96a | 58.77ab | 7.45ab | 19.65a | 10.32b | 9.77a | 43.55a | 0.070b | 0.015a | 0.165ab | 0.758a |
| | 800 | 9.84c | 13.06b | 44.77b | 5.43b | 9.38b | 15.70a | 8.73a | 41.75a | 0.155a | 0.010b | 0.044c | 0.427b |
| | LSD | 40.32 | 26.18 | 15.50 | 2.24 | 5.42 | 3.37 | 2.50 | 14.67 | 0.0518 | 0.0036 | 0.0792 | 0.2258 |
| | mean | 70.51C | 68.74AB | 60.70A | 6.98BC | 16.10A | 10.37B | 9.00A | 37.92B | 0.084C | 0.012C | 0.137B | 0.601B |

(*Continued*)

**Table 1.** (Continued)

| Plant | Salinity (mM NaCl) | Na | Cl | N | P | S | Ca | Mg | K | Fe | Cu | Zn | Mn |
|---|---|---|---|---|---|---|---|---|---|---|---|---|---|
| **Oyster Plant** | 0 | 34.30c | 53.85b | 25.87b | 4.73ab | 7.97a | 18.86a | 10.74a | 44.99a | 0.247a | 0.0089b | 0.086a | 0.552a |
| | 100 | 29.83c | 45.50b | 28.10b | 4.27ab | 7.76a | 19.82a | 9.83a | 35.51ab | 0.236a | 0.0076b | 0.045b | 0.311a |
| | 200 | 103.29a | 197.11a | 31.65b | 4.50ab | 3.35c | 8.59b | 4.64b | 31.52b | 0.176a | 0.0065b | 0.026b | 0.307a |
| | 300 | 93.26a | 86.42ab | 29.97b | 3.42b | 4.45bc | 8.10b | 4.42b | 26.29b | 0.243a | 0.0083b | 0.035b | 0.374a |
| | 400 | 64.10b | 81.13ab | 46.30a | 4.96a | 4.76bc | 9.57b | 4.14b | 34.58ab | 0.276a | 0.0122a | 0.028b | 0.293a |
| | 600 | 69.48b | 71.47ab | 37.93ab | 4.14ab | 5.66b | 9.41b | 4.50b | 29.40b | 0.167a | 0.0090b | 0.031b | 0.362a |
| | LSD | 18.09 | 133.01 | 12.36 | 1.43 | 1.44 | 4.44 | 1.39 | 12.46 | 0.1938 | 0.0028 | 0.0327 | 0.3164 |
| | Mean | 65.71C | 89.25A | 33.30C | 4.34D | 5.66BCD | 12.39A | 6.38B | 33.72B | 0.224B | 0.009D | 0.041E | 0.366C |
| *Anova* | Plant (P) | *** | *** | *** | *** | *** | *** | *** | *** | *** | *** | *** | *** |
| | Salinity (S) | *** | *** | *** | *** | *** | *** | *** | *** | . | *** | *** | NS |
| | P x S | *** | ** | *** | *** | *** | *** | *** | *** | NS | *** | *** | NS |

*Values of mineral concentration are presented as mean of 3 replications. Means that do not share a common lower case letter are significantly different for saline level within a given plant species. Means that do not share a common capital letter are significantly different for the main effect of plant species. Significance codes: 0 '***' 0.001 '**' 0.01 '*' 0.05 '.' 0.1 'NS' 1.

## Experiment 2

**Biomass production.** There was a significant (P < 0.05) effect of NaCl concentration and N source on shoot biomass yield of ice plant and ruby saltbush, and a significant salinity x N source interaction for ruby saltbush biomass yields (Table 2). In both saline and non-saline treatments, shoot biomass of ice plant and ruby saltbush declined with increasing proportions of N supplied as $NH_4^+$ (Table 3). Supply of 100% N as $NH_4^+$ was particularly detrimental, reducing biomass to below 1.25 g/plant (vs as high as 2.28 g/plant with 100% N as $NO_3^-$) in ice plant and below 3.8 g/plant (vs as high as 8.38 g/plant with 100% N as $NO_3^-$) in ruby saltbush. In ice plant, shoot biomass was significantly higher with 200 mM NaCl in the nutrient solution irrespective of N source (Table 3), with a similar response also observed in ruby saltbush (Table 3) Both plants showed a decline in plant weight at 400 mM NaCl.

**Shoot oxalate concentrations.** Shoot oxalate concentrations were significantly affected by N source and salinity in both plant species, and there was a significant N source x salinity interaction in both species (Table 2). In ice plant, shoot oxalate concentrations at all NaCl concentrations declined with increasing proportion of N supplied as $NH_4^+$, from a high of 81.41 (at 400 mM NaCl) with 100% of N provided as $NO_3^-$ to a low of 28.8 mg/g when 100% of N was supplied as $NH_4^+$ (Table 4). A more pronounced decline in shoot oxalate concentrations with

**Table 2. ANOVA table of statistical analysis for the effect of hydroponic solution NaCl concentration and $NO_3$-N: $NH_4$-N ratio (N source) on the shoot biomass, oxalate and nitrate concentration in ice plant and ruby saltbush.**

| Plant | | Shoot biomass | Oxalate concentration | Nitrate concentration |
|---|---|---|---|---|
| Ice plant | Salinity | *** | *** | *** |
| | N source | *** | *** | *** |
| | Salinity x N | NS | * | *** |
| Ruby saltbush | Salinity | *** | *** | *** |
| | N source | *** | *** | *** |
| | Salinity x N | ** | *** | *** |

Significance codes: 0

'***' 0.001 '**' 0.01 '*' 0.05 '.' 0.1 'NS' 1

**Table 3. Effect of hydroponic solution NaCl concentration and NO$_3$-N: NH$_4$+-N ratio on the shoot biomass of ice plant and ruby saltbush.**

| Plant | NO$_3^-$-N: NH$_4^+$-N in nutrient solution | *Shoot DW (g/plant) | | | |
|---|---|---|---|---|---|
| | | NaCl concentration (mM) | | | |
| | | 0 | 200 | 400 | Mean |
| Ice plant | 1.00:0.00 | 1.92±0.44 | 2.28±0.23 | 1.31±0.24 | 1.84A |
| | 0.75:0.25 | 1.59±0.30 | 2.27±0.30 | 1.50±0.23 | 1.79A |
| | 0.50:0.50 | 1.33±0.20 | 2.02±0.28 | 1.17±0.21 | 1.51B |
| | 0.25:0.75 | 1.40±0.29 | 1.81±0.19 | 1.12±0.28 | 1.45B |
| | 0.00:1.00 | 0.84±0.16 | 1.25±0.19 | 0.71±0.11 | 0.93C |
| | Mean | 1.42b | 1.93a | 1.16c | |
| Ruby saltbush | 1.00:0.00 | 7.98±0.39 | 8.38±0.54 | 6.10±0.33 | 7.49A |
| | 0.75:0.25 | 7.39±0.41 | 8.60±0.38 | 5.72±0.18 | 7.24A |
| | 0.50:0.50 | 6.77±0.69 | 7.62±0.90 | 5.90±0.92 | 6.77B |
| | 0.25:0.75 | 6.42±0.66 | 6.43±0.27 | 5.50±0.83 | 6.12C |
| | 0.00:1.00 | 3.79±0.35 | 3.55±0.15 | 2.96±0.31 | 3.43D |
| | Mean | 6.47b | 6.92a | 5.24c | |

The mean shoot dry weight (± standard deviation) of plants treated with different concentrations of NaCl and solutions containing different ratios of NO$_3^-$-N to NH$_4^+$-N is presented for four biological replicates. Italicized letters indicate significant differences between treatments with different NaCl concentrations only or different N source solutions only.

increasing proportions of NH$_4^+$ in the nutrient solution was observed for ruby saltbush. In ice plant, shoot oxalate concentration was similar in the 0 mM control and 200 mM NaCl treatments but was significantly reduced in the 400 mM treatment under all N regimes (Table 4). While there was a significant main effect of salinity on ruby saltbush shoot oxalate concentrations (Table 2), there was no significant pairwise differences between treatment means (range of 5.2–6.9 mg/g; Table 2).

**Table 4. Effect of salinity and nitrogen sources on shoot oxalate concentration in ice plant and ruby saltbush.**

| Plant | NO$_3^-$-N: NH$_4^+$-N in nutrient solution | *Oxalate concentration (mg/g) | | | |
|---|---|---|---|---|---|
| | | Salinity (mM NaCl) | | | |
| | | 0 | 200 | 400 | Mean |
| Ice plant | 1.00:0.00 | 79.78±7.94 | 81.41±8.60 | 63.66±6.22 | 74.95A |
| | 0.75:0.25 | 61.81±8.48 | 70.09±12.18 | 51.23±3.30 | 61.04B |
| | 0.50:0.50 | 54.68±9.91 | 62.93±8.72 | 47.62±0.78 | 55.08C |
| | 0.25:0.75 | 49.50±5.59 | 56.16±4.83 | 45.46±3.06 | 50.37C |
| | 0.00:1.00 | 28.80±2.75 | 37.97±7.35 | 40.75±8.63 | 35.84D |
| | Mean | 54.91b | 61.71a | 49.75c | |
| Ruby saltbush | 1.00:0.00 | 33.00±6.53 | 32.12±5.34 | 50.14±4.22 | 38.42A |
| | 0.75:0.25 | 21.53±3.87 | 27.03±5.82 | 38.37±4.05 | 28.98B |
| | 0.50:0.50 | 10.64±3.26 | 20.25±4.27 | 20.48±3.17 | 17.12C |
| | 0.25:0.75 | 5.73±0.98 | 9.59±1.59 | 7.18±1.89 | 7.50D |
| | 0.00:1.00 | 2.91±0.19 | 2.60±0.22 | 1.20±0.04 | 2.24E |
| | Mean | 14.76c | 18.32b | 23.47a | |

The mean oxalate concentration (± standard deviation) of plants treated with different concentrations of NaCl and solutions containing different ratios of NO$_3^-$-N to NH4$^+$-N is presented for four biological replicates. Italicized letters indicate significant differences between treatments with different NaCl concentrations only or different N source solutions only.

**Table 5. Effect of salinity and nitrogen sources on shoot nitrate concentration in ice plant and ruby saltbush.**

| Plant | $NO_3^-$-N: $NH_4^+$-N in nutrient solution | *Nitrate concentration (mg/g) Salinity (mM NaCl) | | | |
|---|---|---|---|---|---|
| | | **0** | **200** | **400** | *Mean* |
| Ice Plant | 1.00:0.00 | 31.35±6.53 | 46.83±9.07 | 46.39±9.80 | *41.52A* |
| | 0.75:0.25 | 37.38±9.25 | 30.78±5.23 | 29.39±3.17 | *32.52B* |
| | 0.50:0.50 | 28.93±5.53 | 13.00±2.39 | 31.52±4.95 | *24.48C* |
| | 0.25:0.75 | 23.95±2.68 | 5.49±1.24 | 23.12±6.77 | *17.52D* |
| | 0.00:1.00 | 0.48±0.11 | 0.51±0.10 | 0.49±0.06 | *0.49E* |
| | *Mean* | *24.42a* | *19.32b* | *26.18a* | |
| Ruby Saltbush | 1.00:0.00 | 9.94±1.95 | 3.78±1.12 | 3.24±0.43 | *5.65A* |
| | 0.75:0.25 | 8.57±0.51 | 2.49±0.43 | 2.68±0.88 | *4.58B* |
| | 0.50:0.50 | 1.98±0.62 | 1.36±0.20 | 2.17±0.27 | *1.97C* |
| | 0.25:0.75 | 1.81±0.33 | 1.50±0.41 | 2.60±0.51 | *1.84C* |
| | 0.00:1.00 | 1.15±0.36 | 1.49±0.30 | 2.07±0.60 | *1.57C* |
| | *Mean* | *4.69a* | *2.12b* | *2.55b* | |

The mean nitrate concentration (± standard deviation) of plants treated with different concentrations of NaCl and solutions containing different ratios of $NO_3^-$-N to $NH_4^+$-N is presented for four biological replicates. Italicized letters indicate significant differences between treatments with different NaCl concentrations only or different N source solutions only.

**Shoot nitrate concentrations.** There was a significant effect of N source, salinity and their interaction on shoot $NO_3^-$ concentrations in both ice plant and ruby saltbush (S2 Table in S1 File). Despite the significant salinity effect for ice plant there were no significant pairwise differences between treatment means (range of 19.3–6.9 mg/g). In ruby saltbush, shoot $NO_3^-$ concentrations were significantly higher in the 0 mM NaCl control treatment (9.94 mg/g) than the 200 mM and 400 mM NaCl treatments (3.8–3.2 mg/g) (Table 5). Shoot $NO_3^-$ concentrations in ice plant decreased significantly with each increase in the proportion of N supplied as $NH_4^+$ at all NaCl concentrations (Table 5). In ruby saltbush, shoot $NO_3^-$ concentrations were significantly higher with 100% and 75% of N supplied as $NO_3^-$ but then concentrations decreased significantly with a lower proportion of $NO_3^-$ in the nutrient solution (Table 5).

## Discussion

### Effect of salinity on biomass production

By definition, a halophyte is a plant that is able to survive and reproduce in salt concentrations of 200 mM NaCl or more [27]. However, plants classified as halophytes can show a range of responses to salt and differ markedly in the level of salt tolerance [28, 29], as observed in this study. While sea kale and oyster plant grew in salt concentrations up to 800 mM and were therefore 'salt tolerant', their biomass production was highest in the absence of NaCl and declined progressively with increase salt levels. In contrast, ice plant, land seaweed, ruby saltbush and rock samphire increased biomass production with 100 mM of NaCl in the irrigation solution. This growth stimulation by salinity is characteristic of many halophyte plants [30–33], and has been attributed to an increase in turgor brought about by the accumulation of Na ions [34]. While no published data on growth rates of sea kale, oyster plant, land seaweed and ruby saltbush could be found, the growth stimulation in response to low or moderate levels of NaCl has been previously reported for ice plant [4, 7, 28, 35], rock samphire [36, 37], and other halophytes including *Salicornia* species [21, 31, 38, 39] and *Suaeda maritina* [38]. The survival

of all six species in the present study up to 800 mM NaCl suggests they would all be suitable for cultivation as food crops in saline environments.

## Effect of salinity on mineral composition in plants

One of the major effects of salt stress in plants is induced nutritional deficiency; this deficiency may result from the effect of salinity on nutrient availability, competitive uptake and transport, or partitioning within the plant [40–42]. Halophytes have the ability to hyperaccumulate Na in the vacuoles of the cells in the aerial parts of the plant [43–45], and the accumulation of Na can in turn alter the amount of other important macro and micro nutrients [8]. The highest accumulation of Na was found in ice plant and ruby saltbush grown in the presence of NaCl (Table 2), in agreement with concentrations reported in these species in earlier studies [4, 5, 28, 46, 47]. In contrast, land seaweed, rock samphire, sea kale and oyster plant had lower shoot Na concentrations [21, 32, 37, 48]. The largest increase in shoot Na concentration for all species occurred with the addition of 100 mM NaCl to the irrigation solution with the exception of oyster plant. In all species shoot biomass production declined beyond 100 mM or 200 mM NaCl. Therefore, this level of salinity may represent an optimal concentration for maximizing biomass yields while providing a salty flavor for the edible halophytes.

The impact of increased salinity on the concentrations of Ca, Mg, Fe, Zn and Cu has nutritional implications since these nutrients are frequently lacking in the diet of a large proportion of the world's population [49]. Shoot Ca concentrations showed very little change with Na indicating levels of this important nutrient would not be affected if plants were grown in NaCl for commercial production. Calcium is reported to protect membrane structure and function under salt stress [50], and unaltered or increased Ca concentrations have been previously observed in response to increased salinity [35, 46, 51–53]. Notably, however, species differed in shoot Ca concentrations regardless of salinity, with oyster plant having the highest concentrations across all salt treatments. That the addition of NaCl had little impact on the concentrations of Fe, Zn and Cu in the shoots of the edible halophytes, and did not affect Mg concentrations in sea kale and ruby saltbush, indicates that enhancing the salty flavor of edible halophytes with saline irrigation water will not have a negative impact on the nutritional quality for key dietary elements.

## Effect of salinity and $NO_3^-$-N: $NH_4^+$-N ratios on shoot growth and antinutrient concentrations

Increasing the proportion of N provided as $NH_4^+$ led to a reduction in shoot oxalate concentrations in both ice plant and ruby saltbush, while addition of NaCl at 200 mM or 400 mM had no significant effect on shoot oxalate concentrations in either species compared to plants grown in the absence of NaCl. Similar findings have been reported for other halophytes including old man saltbush [14], purslane [22, 24, 26] and warrigal greens [16], where exposure to increasing $NH_4^+$ concentrations significantly decreased leaf oxalate concentrations. These results can be explained because $NO_3^-$ absorbed by roots is subsequently reduced in shoots by the $NO_3^-$ reductase enzyme before the N can be used by the plant [54]. This reaction results in the production and accumulation of organic acids such as oxalic acid in the leaves and stems. Libert and Franceschi [54] also proposed that $NO_3^-$ ions inhibit oxalic acid oxidase activity, preventing the breakdown of oxalic acid, resulting in the accumulation of oxalic acid in the leaves and stems.

Increasing the $NO_3^-$-N: $NH_4^+$-N ratio in the nutrient solution also reduced shoot $NO_3^-$ concentrations in both ice plant and ruby saltbush, but the lower shoot oxalate and $NO_3^-$ concentrations came with concurrent reductions in shoot biomass. The reduction in biomass with an increasing proportion of N supplied as $NH_4^+$ is in agreement with earlier findings for other plant species [14, 16, 26, 55], and may be due to the toxic effects of high tissue concentrations

of $NH_4^+$ observed in some species [22, 56]. High $NH_4^+$ concentrations have been shown to cause damage to photosynthetic machinery, reductions in carbon supply, oxidative stress, changes in cytoplasmic pH, and even hormonal imbalance [56, 57]. Further work would be needed to determine the exact mechanism resulting in reduced biomass under high $NH_4^+$ in the ice plant and ruby saltbush.

## Conclusions

All but two of the six edible halophytes (oyster plant and sea kale) increased biomass production under moderately saline conditions (100 or 200 mM NaCl) and there was no significant reduction in the concentration of key dietary minerals (Cu, Zn, Fe, Ca, and Mg) in any species under saline conditions. The deleterious effect of high levels of NH4+-N on biomass production may be minimized somewhat by salinity treatment, as the addition of 200 mM NaCl was beneficial for biomass production and had no negative impact on either shoot oxalate or $NO_3^-$ concentrations. Therefore, the addition of 100–200 mM salt in irrigation water may be an optimal management practice for greenhouse production of these edible halophytes to increase biomass and maintain concentrations of key dietary minerals without increasing shoot oxalate or $NO_3^-$ concentrations. Given the trade-off between biomass production and minimizing oxalate concentrations in shoots, further work is recommended to examine whether using nitrate nutrition for seedlings to promote biomass accumulation, followed by ammonium nutrition in later growth stages to minimize oxalates, may achieve high yielding crops with low concentrations of anti-nutrients. While this study was conducted in a greenhouse setting, it would be valuable to assess the performance of the halophytes and their viability in soils where traditional crop productivity has declined due to salt accumulation from irrigation. This would help to better assess their potential as alternative crops for areas degraded by soil salinity. This research could provide insights into the feasibility of cultivating these plants in salt-affected soils, thereby offering new options for sustainable food production in regions facing salinity-related agricultural challenges.

## Supporting information

**S1 File.**
(PDF)

## Acknowledgments

Tania Farzana received an RTP scholarship stipend from SCU.

## Author Contributions

**Conceptualization:** Tania Farzana, Terry J. Rose, Bronwyn J. Barkla.

**Formal analysis:** Qi Guo.

**Investigation:** Tania Farzana, Md. Sydur Rahman.

**Methodology:** Tania Farzana.

**Project administration:** Bronwyn J. Barkla.

**Supervision:** Terry J. Rose, Bronwyn J. Barkla.

**Writing – original draft:** Tania Farzana.

**Writing – review & editing:** Qi Guo, Terry J. Rose, Bronwyn J. Barkla.

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
