## [Editor Report · Decision Letter 0]

8 May 2023

PONE-D-23-11250

Salinity and nitrogen source affect productivity and nutritional value of edible halophytes

PLOS ONE

Dear Dr. Barkla,

Thank you for submitting your manuscript to PLOS ONE. After careful consideration, we feel that it has merit but does not fully meet PLOS ONE’s publication criteria as it currently stands. Therefore, we invite you to submit a revised version of the manuscript that addresses the points raised during the review process.

We look forward to receiving your revised manuscript.

Kind regards,

Khandakar R. Islam, Ph.D.

Academic Editor

PLOS ONE

Journal Requirements:

Additional Editor Comments:

The author(s) should perform appropriate statistical analyses (Tables 2 to 4) to improve the quality of the manuscript prior to sending to the reviewers for evaluating its suitability for publication. 

The manuscript in its current format is unsuitable to review due to lack of statistical analysis. Data presented in the Tables 2 - 4 of the manuscript need to be further analyzed using appropriate statistics. Moreover, data presented in the Table 1 were not appropriate. Place the ANOVA information at the bottom of the table to avoid confusion. Once done, the manuscript needs to be resubmitted for review.

---

## [Author Response · Author response to Decision Letter 0]

14 May 2023

As suggested we have revised Table 1 and moved the ANOVA analysis results to Part B of the Figure.

We have also now included statistical analysis for Tables 2 to 4.

We have deleted Supplementary Table 2.

---

## [Decision Letter · Decision Letter 1]

26 Jun 2023

PONE-D-23-11250R1Salinity and nitrogen source affect productivity and nutritional value of edible halophytesPLOS ONE

Dear Dr. Barkla,

Thank you for submitting your manuscript to PLOS ONE. After careful consideration, we feel that it has merit but does not fully meet PLOS ONE’s publication criteria as it currently stands. Therefore, we invite you to submit a revised version of the manuscript that addresses the points raised during the review process.

The authors are advised to revise the manuscript thoroughly especially by addressing the proactive comments of the 1^st^ reviewer. Moreover, authors need to justify why they have conducted two experiments and how these experiments complement each other to achieve the goal of the study? A hypothesis should be added in the introduction to justify accepting or rejecting the findings of the research study. The manuscript needs to be thoroughly edited.  

We look forward to receiving your revised manuscript.

Kind regards,

Khandakar R. Islam, Ph.D.

Academic Editor

PLOS ONE

Journal Requirements:

Reviewers' comments:

Reviewer's Responses to Questions

**Comments to the Author**

1. If the authors have adequately addressed your comments raised in a previous round of review and you feel that this manuscript is now acceptable for publication, you may indicate that here to bypass the “Comments to the Author” section, enter your conflict of interest statement in the “Confidential to Editor” section, and submit your "Accept" recommendation.

Reviewer #1: (No Response)

Reviewer #2: All comments have been addressed

2. Is the manuscript technically sound, and do the data support the conclusions?

Reviewer #1: Partly

Reviewer #2: Yes

3. Has the statistical analysis been performed appropriately and rigorously? 

Reviewer #1: No

Reviewer #2: Yes

4. Have the authors made all data underlying the findings in their manuscript fully available?

Reviewer #1: No

Reviewer #2: Yes

5. Is the manuscript presented in an intelligible fashion and written in standard English?

Reviewer #1: Yes

Reviewer #2: Yes

6. Review Comments to the Author

Reviewer #1: The article titled “Salinity and nitrogen source affect productivity and nutritional value of edible halophytes” by Bronwyn et. al. is a unique piece of research and have merits of its own. However, as per observation of the reviewer, there are some queries about this article along with some decent scopes of modifications which need to be addressed as follows-

1. Please reframe the abstract by adding research methodologies and treatment combinations in brief. Only show key results in the abstract that are conclusive towards research objectives. Please also mention full forms of the abbreviations used, then use abbreviations in the text or abstract.

2. Please also mention in the abstract that there are two sets of experiments in this research, along with their treatment combinations and methodologies.

3. Please point out why there are two experiments in this research? Mention the research objectives clearly at the end of the introduction section (in relevance to the two sets of experiments). Please also mention which experiment fulfils which objective/s and how they interrelate with each other in terms of a conclusive remarks to this research.

4. Please mention the meteorological characteristics of the experimental site/s, along with the primary (before sowing) characteristics of the planting materials and irrigation water, especially in terms of soluble salt contents.

5. Please also mention growth phase duration and timeline for the plants under both of the experiments as well as growth phases and/or point of data collection for both plant and soil.

6. Please explain the rationale to harvest plat shoots at or near flowering stage (in line number 137).

7. Please mention the version of R-software.

8. Please arrange the entire result section objective wise.

9. Elaborate the Discussion section at least to satisfy the objectives. It is highly suggestive that the authors discuss each objective and at the end interrelate them to a conclusive remark.

10. Please explain tables and figures within the text. There are barely some mentions of those in the text.

Reviewer #2: The authors are hereby referred to the reviewer comments attached to this review. I recommend acceptance pending the minor issues highlighted in those comments.

7. PLOS authors have the option to publish the peer review history of their article (what does this mean?). If published, this will include your full peer review and any attached files.

Reviewer #1: **Yes: **Dr. PRAVAT UTPAL ACHARJEE

Reviewer #2: No

---

## [Author Response · Author response to Decision Letter 1]

28 Jun 2023

PLease see attached documents for our response

---

## [Editor Report · Decision Letter 2]

29 Jun 2023

Salinity and nitrogen source affect productivity and nutritional value of edible halophytes

PONE-D-23-11250R2

Dear Dr. Barkla,

We’re pleased to inform you that your manuscript has been judged scientifically suitable for publication and will be formally accepted for publication once it meets all outstanding technical requirements.

Kind regards,

Khandakar R. Islam, Ph.D.

Academic Editor

PLOS ONE
---

## [Editor Report · Acceptance letter]

3 Aug 2023

PONE-D-23-11250R2 

Salinity and nitrogen source affect productivity and nutritional value of edible halophytes 

Dear Dr. Barkla:

I'm pleased to inform you that your manuscript has been deemed suitable for publication in PLOS ONE. Congratulations! Your manuscript is now with our production department. 

Kind regards, 

on behalf of

Dr. Khandakar R. Islam 

Academic Editor

PLOS ONE